# Taking a Breath of the Wild: Are geoscientists more effective than non-geoscientists in determining whether game-world landscapes are realistic?

Rolf Hut[1], Casper Albers[2], Sam Illingworth[3], and Chris Skinner[4]

[1]Delft University of Technology, Delft, The Netherlands
[2]University of Groningen, Groningen, The Netherlands
[3]Manchester Metropolitan University, Manchester, UK
[4]University of Hull, Hull, UK

**Correspondence:** Rolf Hut (r.w.hut@tudelft.nl)

**Abstract.** From the wilderness of Hyrule, the entire continent of Tamriel, to Middle Earth, players of videogames are exposed to wondrous, fantastic, but ultimately fake, landscapes. Given the time people may spend in these worlds, compared to the time they spend being trained in geoscience, we wondered if expert geoscientists would differ from non-geoscientists in whether they judge the landscapes in these games to be "realistic". Since games have a great opportunity for tangential learning it would be a missed opportunity if it turns out that features obviously fake to geoscientists are perceived as plausible by non-geoscientists.

To satisfy our curiosity and answer this question we conducted a survey where we asked people to judge both photos from real landscapes as well as screenshots from the recent *The Legend of Zelda: Breath of the Wild* videogame on how likely they thought the features in the picture were to exist in the real world. Since game-world screenshots are easily identified based on their rendered, pixalated nature, we pre-processed all pictures with an artistic "Van Gogh" filter that removed the rendered nature, but retained the dominant landscape features.

We found that there is a small but significant difference between geoscientists and non-geoscientists with geoscientists being slightly better at judging which pictures are from the real world versus from the game world. While significant the effect is small enough to conclude that fantastical worlds in games can be used for tangential learning on geoscientific subjects.

## 1 Introduction

Modern videogames often provide players with fictional worlds that the players (characters) can explore. While some game worlds include utterly alien (S*tar Wars The Old Republic*, *Horizon Zero Dawn*) or dense urban landscapes (*Grand Theft Auto 5*, *Spiderman*) many offer a world that has large stretches of 'natural environment' as could be found on Earth (*The Legend of Zelda: Breath of the Wild*, *Middle Earth*, *Red Dead Redemption*). However, many of these natural environments contain elements that are, from a geoscientific point of view, unrealistic. This could be either due to the restraints of having to provide

an engaging game, or because the game designers want to present the players with a fantastic, epic setting for their game. The most strikingly unrealistic aspect of many games is that different climate zones are often represented on a relatively small area. For example, the entire world of *Red Dead Redemption 2*, although considered massive for a game, only covers 75 square kilometres (Reddit, 2018), yet includes deserts, prairies, grassy planes, forests and mountain ranges. Similarly, the world of

Hyrule in *The Legend of Zelda: Breath of the Wild* was designed to be "about as big as Kyoto" (Webster, 2017), yet it includes, again, sweltering sand deserts, mountain ranges, swamps and a freezing arctic tundra.

Games have a great potential for tangential learning, i.e. learning things about the real world as a tangential benefit while primarily enjoying the experience (Portnow, 2012; Mozelius et al., 2017). The tangential learning opportunities of videogames have been studied elsewhere (see e.g. Breuer and Bente (2010); Mozelius et al. (2017)); however, what has not yet been fully

addressed is the extent to which this tangential learning could lead to misinformation if the game world was presented in a manner that was incongruent with reality. As such, we wondered if presenting unrealistic geo-features in a game might lead to erroneous learning, i.e. might gamers pick up flawed knowledge of geo-features in our real world because they are presented as realistic within the game world? To test this hypothesis, we conducted a survey in which people were presented with images from the real world and screenshots from games, before being asked to rate how realistic they thought the depicted landscape

was. To make sure that the different images were not recognizable as "from a game" versus "from the real world" (e.g. due to rendering and pixilation), while still depicting the landscapes we wanted study, we used an automated artistic "van Gogh" filter, available at LunaPic.com LunaPic (2015). This filter hides the detailed nature of the image by replacing pixels with brush-strokes, but retains the overall shape of geological features depicted in the image.

Videogames are often reported in the popular press as having supposed negative consequences, such as those associated

with addiction, violence, and isolation (Ferguson, 2007). However, several studies (dating back to the 1980s) have also shown that there are many positive benefits to be gained from playing videogames, such as improving the hand-to-eye coordination, self-esteem, and even the social interactions of the players (see e.g. Griffiths (2002), Granic et al. (2014), Wang et al. (2018)). The educational benefits of playing videogames has also been well studied and documented (Squire (2002), Gee (2003), Squire (2003), Mayer (2019)), and the potential for videogames to contribute towards scientific education is highlighted in the

following quote from Gee (2003) p. 20, who states that:

> Designers face and largely solve an intriguing educational dilemma, one also faced by schools and workplaces: how to get people, often young people, to learn and master something that is long and challenging–and enjoy it, to boot.

As noted by Mayo (2009), videogames have tremendous mass appeal, reaching audiences in the hundreds of thousands to

millions, and so videogames would seem to be an ideal medium through which to communicate geoscientific topics, especially in informal learning environments.

As noted by Dudo et al. (2014), videogames now represent one of the primary platforms through which the general public, and in particular children and adolescents, observe and interact with scientists, and given their global reach and popularity they are fast becoming a key science touch point for average citizens. As well as being an important tool in engaging non-

traditional audiences (Newman et al., 2012), videogames offer the opportunity to spark meaningful and organic engagement around a particular topic (Curtis, 2014). However, if videogames convey information that is incorrect or misleading then it might be that this engagement serves to detract from, rather than contribute towards, the development of meaningful scientific discourse by members of the general public (Squire, 2003). In understanding and constructing meaning from videogames, individuals process the images and elements of design (Rodríguez Estrada and Davis, 2015), and it is the purpose of this study to better understand how this processing enables non-geoscientific audiences to differentiate between realistic and unrealistic geo-features.

In the methods section below we explain the setup of our survey and the statistical methods used to analyse the results. In the results section we present our findings and, in the conclusions, we contextualise these findings and discuss further opportunities for research. Finally, Appendix A contains a post-hoc analyses of the survey data, to look for further interesting patterns. The entire survey is provided as supplementary material.

## 2 Methods

The main question this research seeks to answer is: "do people without a background in the geosciences perceive landscapes from game worlds as more realistic compared to those with a background in the geosciences?" To study this question, we took six images from the game *The Legend of Zelda: Breath of the Wild* (*BotW*). The images were chosen to represent a wide variety of landscapes (i.e. a volcano, a tropical forest, a grassy plane, etc.) and needed to include a geological features as central criteria. The images were chosen such that no clear landmarks that identified it as a fantasy game, such as iconic temples, towers, or Hyrule Castle were visible in the picture. To select the images used in this study we constructed a list of landscape types (volcano, arctic, desert, plains, swamps, jungle) that we wished to include in the survey. Author Hut (who has an intimate knowledge of the gameworld) selected six locations that did not include any recognizable features and made screenshots using the in-game camera option. Each image was used as input in a reversed image search in the Google search engine. From the search result real world images were hand picked. To determine if participants could distinguish made up landscapes in games from real landscapes, for each picture taken from *BotW* an accompanying image from the real world was sought by doing a a reverse image search. Since the *BotW* screenshots can be easily recognized due to their rendered nature, we choose to use an automated artistic filter on both the *BotW* screenshots and on the real world images. The filter needed to alter both type of images enough not to be able to distinguish them based on the rendering nature of the screenshots, yet retain the essential geo-features of the landscape. To make sure that all of the images were treated equally we needed a filter that works algorithmically and does not require input from a human. Many such filters exists online, yet most of them either failed in removing the rendered nature of the videogame images, or else modified the image too much, removing any the geoscientific content. The "van Gogh" filter (available through LunaPic (2015)) was chosen as it was one of the few filters which retained the geological features of the image, while hiding the rendered nature of the videogame images. Figure 1 shows how two images (*BotW* screenshot on the upper left and real world photo on the upper right) were transformed using the filter. Both the original as well as the filtered figures are available in the supplementary material.

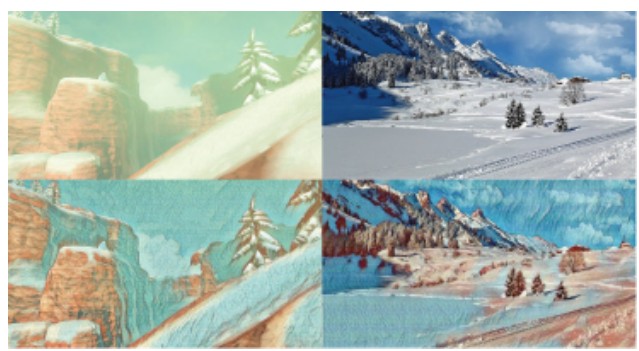

**Figure 1.** Two images used in the survey. The top two images are original and the bottom two images are processed through the "van Gogh" filter. The left two images are from the game *The Legend of Zelda: Breath of the Wild* and the right two images are from the real world. The bottom two figures where presented in the survey with the question: "Knowing that this picture has gone through a "van Gogh"-filter, how likely do you think it is that the features depicted in the artistic renderings could exist in the real world? We use a 10 point scale where 1 = completely unlikely to 10 = completely likely". High-resolution versions of images (originals and after filters were applied) are available in the supplementary material.

## 3 Survey design

To test our hypothesis, we wanted to know if people are capable of recognizing fake landscapes from games and if geoscientists are better at this than non-geoscientists. When asking an assessment on whether someone thinks a certain photo is real or faked, there are basically two options to do so. One option is to ask "do you think this photo is real" via a yes/no question; the other is to ask "How certain are you that this photo is real" on an ordinal scale. The benefit of such ordinal scales is that they can incorporate more nuance than a simple dichotomy, as one can also indicate that one isn't sure. Such likert scales are extremely common, especially within the social sciences. Common scales are 5, 7 and 9-point scales. See Lord and Novick (2008) or Kaplan and Saccuzzo (2017) for more background on questionnaire design. Therefore, for all the pictures generated as described above we asked the question: "Knowing that this picture has gone through a "van Gogh"-filter, how likely do you think it is that the features depicted in the artistic renderings could exist in the real world? We use a 10 point scale where 1 = completely unlikely to 10 = completely likely"

To distinguish between people with and without a background in the geosciences, we asked the question "Do you consider yourself a geoscientist?"

We wanted to exclude participants who had previously played *BotW*, as they could potentially recognize places from the game, skewing the results. At the same time, we did not want to alert everyone to the fact that the pictures they were looking at were taken from this particular game, therefore we added a broad question asking participants which games they had played in the last year, which included *BotW*.

As additional background information to be able to do post-hoc analyses we added questions on age, gender and highest completed education level. The entire survey, including the required legal statements on voluntary participation, proper handling of private information and the option to quit at any time, is provided in the supplementary material.

The survey was carried out using two methods: printed version of the survey were handed out at the European Geoscience Union (EGU) General Assembly 2018 in Vienna. In this way we intended to reach people with a background in the geosciences. After the assembly, an online version of the survey, designed in Google Forms and available through rolfhut.nl/botw (Hut, 2018) was announced using social media of the authors during the conference and in the week afterwards (April 8 through April 20, 2018). Advertising a survey through social media runs the risk of selecting survey participants from the biased social circle of the authors. However, Côté and Darling (2018) showed that above one thousand followers, a more diverse public is reached through twitter. Given that all the authors shared the survey through their personal twitter accounts, and that each of these accounts has more than 2,000 followers we are confident that an acceptably diverse public was reached using this approach.

## 4  Statistical analyses

All 12 pictures, 6 Zelda-pictures and 6 real pictures, were rated by all participants on a 1–10 scale. The rating on photo $i$ by participant $j$ is denoted by $r_{i,j}$. As a first step, we study on a picture-by-picture basis whether the mean ratings of geoscientists differ from that of laypeople. For this, we use Student's $t$-test with Bonferroni-correction to account for multiple testing.

However, our main interest is not in the individual pictures, but in the overall message from the 12 pictures. An overall penalty score per person is constructed. For each photo, participants have rated the photos on a scale from 1 (=fake) to 10 (=real). The best answer they can give is 1 for the 6 game world photos and 10 for the 6 real world ones. For each photo, the distance between the given answer and the best answer is calculated (thus 0 when the participant is fully correct up to 9 when (s)he's fully wrong). The absolute value of the 12 distance-scores for the photos are added. The resulting metric is a penalty as a low score is good (the perfect candidate scores 0, one who's as wrong as it gets scores 12*9=108).

To operationalize this we define $K = (10, 1, 1, 10, 10, 1, 1, 10, 10, 1, 1, 10)$ as the vector of ideal answers ('1' for each picture from *BotW* and '10' for the real ones). We compute the score by for each person in the survey simply counting the sum of the distances to this optimal answer:

$$\text{Penalty}_i = \sum_{j=1}^{12} |r_{i,j} - K_i|. \tag{1}$$

The score patterns of geoscientists and non-geoscientists will be compared both visually as through a Student $t$-test.

All hypothesis tests have been carried out two-sided. The code to run these analyses as well as the anonymized survey files are available at Albers and Hut (2019).

**Table 1.** Average rating and SD per picture for geoscientists and non-geoscientists. The first six rows concern real pictures, the last six BotW-pictures.

|  | Geoscientists | | Laypeople | |
|---|---|---|---|---|
|  | Mean | SD | Mean | SD |
| Picture.1 | 7.952 | 2.409 | 7.655 | 2.453 |
| Picture.4 | 9.476 | 0.814 | 8.879 | 1.836 |
| Picture.5 | 9.071 | 1.438 | 8.690 | 1.749 |
| Picture.8 | 8.119 | 2.515 | 7.776 | 2.435 |
| Picture.9 | 9.131 | 1.387 | 8.638 | 1.861 |
| Picture.12 | 7.726 | 2.481 | 7.603 | 2.554 |
| Real pictures | 8.579 | 1.112 | 8.207 | 1.164 |
| Picture.2 | 3.702 | 2.104 | 5.017 | 2.544 |
| Picture.3 | 6.274 | 2.500 | 6.621 | 2.553 |
| Picture.6 | 6.381 | 2.737 | 6.086 | 2.529 |
| Picture.7 | 5.798 | 2.419 | 5.948 | 2.228 |
| Picture.10 | 5.940 | 2.485 | 6.655 | 2.482 |
| Picture.11 | 6.381 | 2.755 | 6.224 | 2.596 |
| BotW pictures | 5.746 | 1.685 | 6.092 | 1.566 |

## 5 Results

A total of 163 people completed the survey. Four participants did not score all pictures and were excluded from the data. Furthermore, 17 participants indicated that they had played *BotW* and were also excluded. Of the remaining 142 participants, 84 (59%) of them indicated that they were a geoscientist. The average rating and standard deviation of the ratings for geoscientists and non-geoscientists is provided in Table 1. Full aggregated scores per picture per group (geoscientist versus non-geoscientist) are provided in the Appendix (Tables A1 and A2). Figure 2 shows the distribution of penalty-scores for both groups. From the visual comparison, we observe that geoscientists, on average, have lower penalty scores than non-geoscientists. Also from the table, we see that compared to the non-geoscientists the geoscientists gave higher scores for the real world pictures and lower scores for the in game screenshots, indicating that they are better at telling the difference. Since our research only tests if people recognize video game world images as not realistic, we can not say through what mechanism geoscientists arrive at their better score compared to non-geoscientists. We hypothesize that this could be an effect of their training, or an effect of being exposed to many real geoscientific images during their career. Concluding that either providing more training, or more exposure to geoscientific images, to non-geoscientists to improve their ability to recognize non realistic landscapes in video

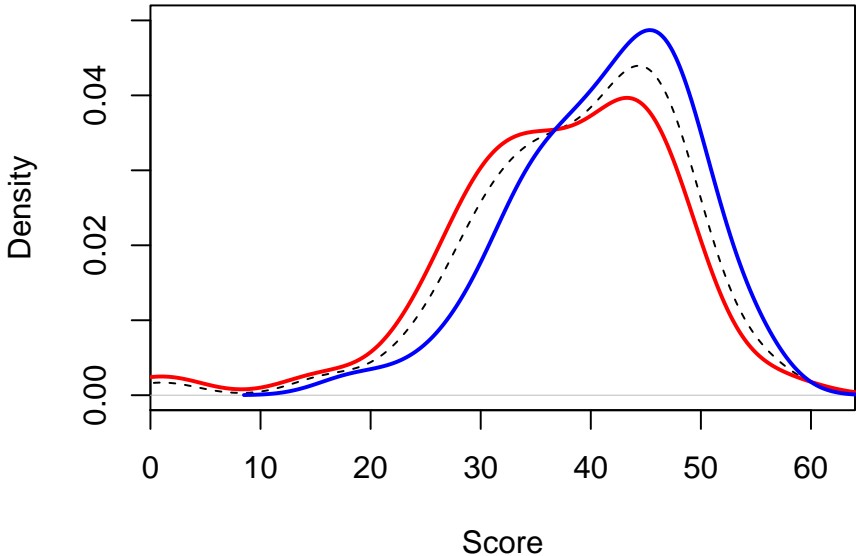

**Figure 2.** Distribution of the penalty-scores. The red curve denotes the geoscientists, the blue one the non-geoscientists. The dashed black curve shows the density for the whole sample.

games is not possible based on these results. A more qualitatively focused follow up research could potentially unearth this reason.

On an individual picture level, only the difference for Picture 2 is statistically significant ($t = 3.358$, $df = 140$, $p_{Bonf} = .012$). This hints that the difference between geoscientists and non-geoscientists is most likely small.

Statistically testing our overall hypothesis, thus combining the information over all 12 pictures, we find that geoscientists indeed score better at separating fake landscapes from real ones ($t = 2.704$, $df = 140$, $p = .008$). However, the effect size is rather small: the mean combined penalty in the laypeople group is 41.31, and in the geoscientist group is 37.00. Thus, geoscientists perform about 10% better, according to the metric in Equation (1). When including additional covariates (whether or not someone is a gamer, gender, and age) in the analyses, the message remains that geoscientists perform about 10% better, i.e. even when correcting for those covariates the difference between geoscientists and non-geoscientists remains significant (see Table A3). Out of the 84 experts, 34 filled in the questionnaire on paper and the rest digitally. Of the 58 non-geoscientists only 4 filled in the questionnaire on paper. Different modes of administration for both groups, could have had a small effect on the outcomes.

Other interesting patterns are visible in the collected data. For example, (table A3) the effect of being a self-identified gamer have no influence on the result. This surprises us, since we would expect that people being exposed more to games might have a better sense for which landscapes are from a game. We have to stress that this is a post-hoc analyses that only results in a hypothesis that needs further testing.

## 6    Conclusions

We have demonstrated that while geoscientists might be slightly, but statistically significantly, better at separating real world photos of landscapes from game screenshots, non-geoscientists are still capable of identifying landscapes from a game, even when both the real world photos and the game screenshots are filtered through an artistic 'van Gogh' filter. This suggests that people recognize the natural features in game worlds for the fantastical settings that they are. On a 10 point scale lay people rate real world images on average 2.115 points higher than images from the videogame. Geoscientists rate real world images 2.833 points higher. Whether people are able to distinguish factual information (a volcano is hot) from erroneous information (an arctic tundra within a five-minute walk from a sweltering desert) within the context of a game is a question that requires further research.

Care needs to be taken when interpreting the results of this study. While we clearly asked people to indicate if the features in the picture could exist in the real world, and we do not ask to judge if the picture is from a game, there is a chance that some people answered the question with this in mind. To prevent this, we used the "van Gogh" filter to hide the fact that the game screenshots were from a rendered video game image. However, people with experience in playing games might still look for tell-tale signs of video game-generated images. Landscapes in games are, of course, designed by artists who draw on the real world for inspiration. Different games will have different levels of adherence to real world inspirations, depending on the aesthetic sensibilities of the gameworld or limitations of the software used to create it. Our research focused on the question: can non-geoscientific experts correctly identify non-realistic aspects of landscapes in videogames. The paper survey was handed out at the EGU General Assembly 2018, to target geoscientists. However, this was done during the "games in geoscience"-session, potentially over representing gamers among geoscientists. Post-hoc analyses showed no significant over representation of gamers among geoscientist (see table A3). More detailed qualitative research, guided by the results of this research, could further test the assumptions made in this work, and provide insight into the manner by which people determine whether an image is real or not. Finally, the distinction between geoscientists and non-geoscientists is made based on an acknowledgment of formal education in the geosciences. In the non-geoscientists group there is potentially a large difference between people that do, and do not, get exposed to geological features; for example, through hiking, watching documentaries, etc. Follow up research, either qualitative or quantitative, could include questions on self-perceived level of geoscientific knowledge. Doing so would likely yield further information on how well versed those educated in the geosciences consider themselves, versus those not educated, perhaps demonstrating a geoscientific version of the Dunning-Kruger effect (Dunning, 2011).

We have shown that even though the difference in the ability to identify whether an image is from a videogame, or from the real world is significant, the effect size is small and the overall scores are high. Whilst further study is needed to fully assess the

effectiveness of videogames when used in this manner, this study indicates that wrongfully interpreting game world landscapes as real is not a risk when aiming to tangentially communicate geoscientific principles.

*Code and data availability.* The anonymized survey results, as well as the code that was used to perform the statistical analyses, is available through Albers and Hut (2019)

## 5 Appendix A: Additional tables

**Table A1.** Frequencies of ratings by geoscientists for all pictures. The first six columns are the real pictures, the last six columns the BotW-pictures.

|  | P1 | P4 | P5 | P8 | P9 | P12 | P2 | P3 | P6 | P7 | P10 | P11 | $\sum$ |
|---|---|---|---|---|---|---|---|---|---|---|---|---|---|
| Grade: 1 | 1 | 0 | 1 | 3 | 0 | 3 | 13 | 2 | 2 | 4 | 2 | 3 | 34 |
| Grade: 2 | 1 | 0 | 0 | 3 | 0 | 2 | 12 | 5 | 3 | 2 | 5 | 2 | 35 |
| Grade: 3 | 5 | 0 | 0 | 2 | 0 | 4 | 22 | 8 | 9 | 9 | 7 | 11 | 77 |
| Grade: 4 | 5 | 0 | 0 | 2 | 1 | 1 | 12 | 6 | 12 | 11 | 13 | 13 | 76 |
| Grade: 5 | 3 | 0 | 1 | 1 | 2 | 5 | 9 | 12 | 13 | 18 | 15 | 5 | 84 |
| Grade: 6 | 5 | 0 | 2 | 5 | 3 | 3 | 4 | 8 | 5 | 6 | 7 | 6 | 54 |
| Grade: 7 | 5 | 3 | 4 | 5 | 4 | 11 | 7 | 12 | 6 | 9 | 8 | 9 | 83 |
| Grade: 8 | 12 | 8 | 13 | 13 | 10 | 13 | 4 | 12 | 6 | 11 | 8 | 10 | 120 |
| Grade: 9 | 16 | 19 | 18 | 15 | 13 | 18 | 0 | 11 | 12 | 9 | 13 | 10 | 154 |
| Grade: 10 | 31 | 54 | 45 | 35 | 51 | 24 | 1 | 8 | 16 | 5 | 6 | 15 | 291 |

*Author contributions.* We are using the CASRAI Credit system (Allen et al., 2014) to acknowledge the different roles of the different authors.

– Rolf Hut: Conceptualization, Data curation, Investigation, Methodology, Visualization, Writing – original draft, Writing – review & editing.

– Casper Albers: Methodology, Writing – original draft (statistical analyses and results), Formal analysis, Writing – review & editing.

– Sam Illingworth: Conceptualization, Writing – original draft (introduction), Writing – review & editing.

– Chris Skinner: Conceptualization, Writing – review & editing.

*Competing interests.* None. The authors explicitly state that they have no commercial ties to the Nintendo corporation, producer of the 'Zelda: Breath of the Wild' video game

**Table A2.** Frequencies of ratings by non-geoscientists for all pictures. The first six columns are the real pictures, the last six columns the BotW-pictures.

| | P1 | P4 | P5 | P8 | P9 | P12 | P2 | P3 | P6 | P7 | P10 | P11 | $\sum$ |
|---|---|---|---|---|---|---|---|---|---|---|---|---|---|
| Grade: 1 | 1 | 0 | 0 | 1 | 0 | 0 | 5 | 1 | 2 | 0 | 1 | 1 | 12 |
| Grade: 2 | 0 | 0 | 0 | 1 | 2 | 4 | 3 | 2 | 4 | 3 | 2 | 4 | 25 |
| Grade: 3 | 6 | 3 | 1 | 4 | 0 | 0 | 13 | 5 | 5 | 7 | 6 | 6 | 56 |
| Grade: 4 | 4 | 1 | 2 | 3 | 0 | 5 | 6 | 8 | 6 | 8 | 5 | 7 | 55 |
| Grade: 5 | 0 | 0 | 2 | 1 | 2 | 6 | 6 | 5 | 5 | 7 | 4 | 5 | 43 |
| Grade: 6 | 2 | 1 | 1 | 3 | 3 | 2 | 8 | 2 | 7 | 8 | 7 | 7 | 51 |
| Grade: 7 | 5 | 3 | 4 | 5 | 2 | 6 | 7 | 11 | 12 | 7 | 6 | 6 | 74 |
| Grade: 8 | 14 | 7 | 10 | 12 | 13 | 6 | 4 | 7 | 6 | 10 | 10 | 7 | 106 |
| Grade: 9 | 10 | 11 | 11 | 10 | 9 | 9 | 2 | 8 | 5 | 6 | 11 | 9 | 101 |
| Grade: 10 | 16 | 32 | 27 | 18 | 27 | 20 | 4 | 9 | 6 | 2 | 6 | 6 | 173 |

**Table A3.** Results of the linear regression model predicting penalty scores from dichotomous variables geoscientist, gamer and male and continuous variable age. Geoscientists and men score significantly better than non-geoscientists and women, respectively. The effect of gamer and age is non-significant.

| | Estimate | SE | $t$-value | $p$-value |
|---|---|---|---|---|
| (Intercept) | 39.505 | 3.360 | 11.76 | <0.001 |
| Geoscientist | -4.125 | 1.653 | -2.50 | 0.014 |
| Gamer | 2.048 | 1.66 | 1.23 | 0.219 |
| Male | -3.499 | 1.63 | -2.15 | 0.034 |
| Age | 0.087 | 0.077 | 1.13 | 0.259 |

*Acknowledgements.* The authors would like to thank all people that have participated in the survey. No funding was received to carry out this research.

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
