# Peer review of "Taking a Breath of the Wild: Are geoscientists more effective than non-geoscientists in determining whether game-world landscapes are realistic?"

_Geoscience Communication, 2019_

## Referee Comment (RC1) · Anonymous Referee #1 · 11 Apr 2019

The paper opens a new field in GC, it is an interesting connection to the currently relevant topic of gaming and to science communication. It addresses the difference between experts and layman.

The scientific method is valid, however I would recommend the authors to incorporate more literature in the introduction. The authors can explore the field of learning in games further. Furthermore, the authors could more explicitly motivate the choice of method, including the choice of these particular images as well as the chosen filter.

[Figure]

The results are sufficient as a first exploration in this research direction, as the authors state in the conclusions. They could focus their conclusion also on the ability to recognise realistic landscapes for both groups.

The conclusion that this study indicates the potential to use games as powerful tool to communicate geoscientific principles is unclear to me.

The presentation is well structured, the title reflects the content, the language is fluent and precise. The number and quality of references can be improved by a stronger introduction of the research and method.

---

## Referee Comment (RC2) · Anonymous Referee #2 · 11 Apr 2019

General comments:

Overall, this paper addresses some of the lack of evidence to support the increasing use of video games in geoscience education. The paper is well presented and goes some way to help build the evidence base for the use of games whilst adopting an innovative and interesting approach.

Although the approach adopted is valid for this type of research, the data collection methods are rather light touch and a little disappointing. There is little to no justification

for the data collection methods used and I would like to see more explanations on the relevance of this research to the wider field. This study could also greatly benefit from additional data collections – particularly qualitative data to validate some of the assumptions made within the paper and to really make a significant contribution to the field.

The results show a very marginal difference between the participant groups and some of the conclusions based on these results are a little far-fetched. There is data of real interest here, so try framing it in the way that shows its worth rather than stretching its meaning!

Overall, the paper is well structured, fluent and easy to read. There are a few grammatical errors throughout, some of which are highlighted below. The title of the paper matches the content well.

Specific comments:

The paper is very light on references and could benefit from a more robust background across gaming literature (e.g. there is a wealth of literature on 'stealth learning').

Why the 'van Gogh' filter specifically? Were other filters explored? why were they excluded? Much more detail needs to be added to explain the reasons why and the advantages/bias of using this filter over other filters.

P2 Line 7-8 "Games have a great potential for tangential learning, i.e. learning things about the real world as a tangential benefit while primarily enjoying the experience" – you state this and then immediately try to link to the potential problems (erroneous learning), but why consider the problems and not the potential benefits you reference as possible? Needs justification.

How could you be sure that the 'fake' landscapes from the game had not been designed based on existing real-world landscapes (as surely many are)? You may not be able to evidence this, but this could be a bias in the study and should be acknowledged.

**[GCD](GCD)**

Interactive
comment

As acknowledged by the authors, this research is in desperate need of the addition of qualitative analysis. This would also enable cognitive testing for the use of the filter and how that the use of the filter adds bias to the results, as well as supporting any hypotheses you are trying to prove/disprove.

P3 – why was a scaled-type question used to gather data over other types of questions? Purely for quantitative analysis? Needs justification and literature examples where possible to justify the method.

You use the differences between geoscientists with formal training and laypeople – how do you account for other bias e.g. a layperson who travels a lot or enjoys hiking in nature and is therefore more exposed to the different types of landscapes compared to somebody who spends a lot of time playing videogames inside (with the exception of BotW).

P7 – How were the images distributed at the EGU? By paper on digitally? This could affect the perception of the images compared to the laypersons who saw them digitally. Simple but should be included.

Technical corrections:

P2 Line 14 – citation error. P2 Line 20-23 "However, if videogame. . ." – this sentence needs a citation. P2 Line 30 – perhaps change rate to perceive P3 Figure 1 – needs to be much larger to enable the reader to compare the quality of images. P4 Line 29-30 – this sentence is confusing – overall N=163 of which 4 and 17 participants were removed? It reads more like 163 people only completed part of the survey and were also removed! "filled out part of the survey" – maybe simply change this to 'completed the survey'. P5 Line 6-8 – as mentioned above this could be further qualified if a qualitative approach is adopted.

---

## Short Comment (SC1) · 15 Apr 2019

This is a very interesting paper on whether game environments provoke a type of 'uncanny valley' effect in their depiction of different types of landscapes and whether trained geoscientists are more effective at discerning this compared to the general public. Given that game environments are being more and more widely used to communicate geoscience, this is very relevant.

I have one main comment on the analysis method used and its interpretation. The

numerical scale participants were asked to use essentially measures two potentially independent things: whether participants identify if the picture was real (6-10) or fake (1-5), and how confident they are in that choice (the absolute distance of their score from 5.5). The penalty score used in the paper therefore is measuring a combination of both these effects. While this is appropriate for the question the authors raise of "do people without a background in the geosciences rate landscapes from game worlds as more realistic compared to those with a background in the geosciences?", it is important to understand the underlying factors between the two sets of penalty scores. I would suggest the authors therefore also compare whether there are statistically significant differences in both the success rate and the confidence of geoscientists compared to non-geoscientists and whether these are independent or not. This would then determine if geoscientists are more effective or just more confident (or some combined effect) in their judgements.

In addition, it would be helpful if the authors made a short comment about the likely types of non-geoscientists they were reaching by choosing to share through their own social media channels. Recent research has highlighted the critical number of Twitter followers to start reaching outside of the scientific community [Cote and Darling, 2018, Facets, https://doi.org/10.1139/facets-2018-0002]. Beyond the findings of that study, given that the authors are geoscientists, is it not likely that the people who follow them at least have an interest in the geosciences and therefore may be performing better than a truly representative random sample of the population? If the authors have any evidence of influencers unaffiliated with geoscience who shared the survey, this would also be highly relevant to this point.

---

## Author Comment (AC1) · 6 Jun 2019

**Reply to comments by reviewers**

We thank the reviewers for their thorough reading of our work and their constructive feedback, which we believe will further improve our work. We are happy to read that the reviewers agree with us that "*The paper opens a new field in GC, it is an interesting connection to the currently relevant topic of gaming and to science communication.*" [reviewer #1].

Below we have copied all comments by the reviewers and provided an answer to each comment. For ease of reading, we have written all of the comments from reviewers in *italic*, with all our responses written in normal font. We use P2L5 to indicate page two, line five of the original manuscript. Any suggestion for text to be added to a revised manuscript is indented.

**Anonymous Referee #1**

*The paper opens a new field in GC, it is an interesting connection to the currently relevant topic of gaming and to science communication. It addresses the difference between experts and layman.*

We thank the reviewer for the complements.

*The scientific method is valid, however I would recommend the authors to incorporate more literature in the introduction. The authors can explore the field of learning in games further.*

We agree in hindsight that the introduction is a bit light on references, especially with respect to videogames and learning. Reviewer #2 also pointed this out. We suggest to add the following paragraph after P2L15, which now includes more relevant literature on videogames and learning:

> Videogames are often reported in the popular press as having supposed negative consequences, such as those associated with addiction, violence, and isolation (Ferguson, 2007). However, several studies (dating back to the 1980s) have also shown that there are many positive benefits to be gained from playing videogames, such as improving the hand-to-eye coordination, self-esteem, and even the social interactions of the players (see e.g. Griffiths, 2002, Granic et al., 2014, Wang et al., 2018). The educational benefits of playing videogames has also been well studied and documented (Squire, 2002, Gee, 2003, Squire, 2003, Mayer, 2019), and the potential for videogames to contribute towards scientific education is highlighted in the following quote from Gee (2003, p. 20), who states that:

> Designers face and largely solve an intriguing educational dilemma, one also faced by schools and workplaces: how to get people, often young people, to learn and master something that is long and challenging--and enjoy it, to boot.

As noted by Mayo (2009), videogames have tremendous mass appeal, reaching audiences in the hundreds of thousands to millions, and so videogames would seem to be an ideal medium through which to communicate geoscientific topics, especially in informal learning environments.

Ferguson, C. J. 2007. The good, the bad and the ugly: A meta-analytic review of positive and negative effects of violent video games. *Psychiatric quarterly,* 78**,** 309-316.

Gee, J. P. 2003. What video games have to teach us about learning and literacy. *Computers in Entertainment (CIE),* 1**,** 20-20.

Granic, I., Lobel, A. & Engels, R. C. 2014. The benefits of playing video games. *American psychologist,* 69**,** 66.

Griffiths, M. D. 2002. The educational benefits of videogames. *Education and health,* 20**,** 47-51.

Mayer, R. E. 2019. Computer games in education. *Annual review of psychology,* 70**,** 531-549.

Mayo, M. J. 2009. Video games: A route to large-scale STEM education? *Science,* 323**,** 79-82.

Squire, K. 2002. Cultural framing of computer/video games. *Game studies,* 2**,** 1-13.

Squire, K. 2003. Video games in education. *Int. J. Intell. Games & Simulation,* 2**,** 49-62.

Wang, B., Taylor, L. & Sun, Q. 2018. Families that play together stay together: Investigating family bonding through video games. *New Media & Society,* 20**,** 4074-4094.

*Furthermore, the authors could more explicitly motivate the choice of method, including the choice of these particular images as well as the chosen filter.*

We choose the images by formulating a few constraints: we needed to make sure that no features that were clearly from the game world were present in the image; we needed to cover a wide range of typical (real world) landscapes; and we needed to make sure that at least some geological features were presented in the image. With these constraints in mind author Hut (who has intimate knowledge of the gameworld of *Breath of the Wild*) selected the locations and made images using the in-game screen capture tool. The corresponding real-world examples were chosen using a reverse image search in Google, where the image from the game was used as source to find the corresponding images.

For the filters we briefly considered hiring professional artists to make artistic renditions of the landscape, but realized that the personal influences of the artists on the results would be a problem. Many filters exist online that can procedurally convert an image to an "artsy" version of that image, without personal prejudice. We settled on the van Gogh filter because of all the filters considered it retained the geological features of the image, while hiding the rendered

nature of the videogame images. We suggest adding the following paragraph to the methods section, after P3L1:

> To select the images used in this study we constructed a list of landscape types (volcano, arctic, desert, plains, swamps, jungle) that we wished to include in the survey. Author Hut (who has an intimate knowledge of the gameworld) selected six locations that did not include any recognizable features and made screenshots using the in-game camera option. Each image was used as input in a reversed image search in the Google search engine. From the search result real world images were hand picked.

And after P3L6 change "We choose to use the "van Gogh" filter available online (LunaPic, 2015)." to:

> To make sure that all of the images were treated equally we needed a filter that works algorithmically and does not require input from a human. Many such filters exists online, yet most of them either failed in removing the rendered nature of the videogame images, or else modified the image too much, removing any the geoscientific content. The "van Gogh" filter (available through LunaPic (2015)) was chosen as it was one of the few filters which retained the geological features of the image, while hiding the rendered nature of the videogame images.

*The results are sufficient as a first exploration in this research direction, as the authors state in the conclusions. They could focus their conclusion also on the ability to recognise realistic landscapes for both groups.*

Although we set out to study the difference between lay-people and experts, we agree that the posteriori result that people in general are good in recognizing real landscapes is interesting. We already state in P6L9 that: "This suggests that people recognize the natural features in game worlds for the fantastical settings that they are." However, to further emphasis this result, we suggest adding the following sentence directly after it:

> On a 10 point scale lay people rate real world images on average 2.115 points higher than images from the videogame. Geoscientists rate real world images 2.115 points higher.

Furthermore, we've added the following two lines to Table 1:

| | Geoscientists | | Non-geoscientists | |
|---|---|---|---|---|
| | Mean | SD | Mean | SD |
| Real photos | 8.579 | 1.112 | 8.207 | 1.164 |

| Fake photos | 5.746 | 1.685 | 6.092 | 1.566 |

*The conclusion that this study indicates the potential to use games as powerful tool to communicate geoscientific principles is unclear to me.*

We agree that the claim as it is stated now is a bit too strong based on the research that we have done. Reviewer #2 also mentioned this. We suggest to change "this study indicates that games could potentially be used as a powerful tool through which to tangentially communicate geoscientific principles." to:

> this study indicates that wrongfully interpreting game world landscapes as real is not a risk when aiming to tangentially communicate geoscientific principles.

*The presentation is well structured, the title reflects the content, the language is fluent and precise. The number and quality of references can be improved by a stronger introduction of the research and method.*

See the comments above on suggestions to improve the introduction and methods section, especially with respect to the number and quality of references.

**Anonymous Referee #2**

*General comments: Overall, this paper addresses some of the lack of evidence to support the increasing use of video games in geoscience education. The paper is well presented and goes some way to help build the evidence base for the use of games whilst adopting an innovative and interesting approach.*

*Although the approach adopted is valid for this type of research, the data collection methods are rather light touch and a little disappointing. There is little to no justification for the data collection methods used and I would like to see more explanations on the relevance of this research to the wider field. This study could also greatly benefit from additional data collections – particularly qualitative data to validate some of the assumptions made within the paper and to really make a significant contribution to the field.*

We thank the reviewer for those comments. We agree that this research is a first quantitative exploration that would benefit from follow up qualitative research. We suggest to change "More detailed qualitative research, guided by the results of this research, could shed light on this." on P7L8 to:

More detailed qualitative research, guided by the results of this research, could further test the assumptions made in this work, and provide insight into the manner by which people determine whether an image is real or not.

*The results show a very marginal difference between the participant groups and some of the conclusions based on these results are a little far-fetched. There is data of real interest here, so try framing it in the way that shows its worth rather than stretching its meaning!*

We agree that there is valuable data in the results of our survey. We originally set out to investigate the difference between self-identified geoscientists and non-geoscientists, and as such that is the main focus of the manuscript. Our in-house statistician (author Albers) warns that looking for patterns that were not originally part of the study design constitutes post-hoc analyses and that doing so should always be considered as newly formed hypotheses that need further testing. To emphasize this, we suggest adding the following paragraph at P6L4:

> Other interesting patterns are visible in the collected data. For example, (table A3) the effect of being a self-identified gamer have no influence on the result. This surprises us, since we would expect that people being exposed more to games might have a better sense for which landscapes are from a game. We have to stress that this is a post-hoc analyses that only results in a hypothesis that needs further testing.

Furthermore, we wish to stress that we already tried to be cautious in our wording. For instance, on P5L15 we immediately after showing our significant result stress that the effect size is rather small. We fully agree that results should not be over-interpreted.

*Overall, the paper is well structured, fluent and easy to read. There are a few grammatical errors throughout, some of which are highlighted below. The title of the paper matches the content well.*

*Specific comments:*

*The paper is very light on references and could benefit from a more robust background across gaming literature (e.g. there is a wealth of literature on 'stealth learning').*

We agree in hindsight that the introduction is a bit light on references on learning. Reviewer #1 also pointed this out. As mentioned above and repeated here for convenience, we suggest to add the following paragraph after P2L15, which now includes more relevant literature on learning in videogames:

> Videogames are often reported in the popular press as having supposed negative consequences, such as those associated with addiction, violence, and isolation (Ferguson, 2007). However, several studies (dating back to the 1980s) have also shown that there are many positive benefits to be gained from playing videogames, such as

improving the hand-to-eye coordination, self-esteem, and even the social interactions of the players (see e.g. Griffiths, 2002, Granic et al., 2014, Wang et al., 2018). The educational benefits of playing videogames has also been well studied and documented (Squire, 2002, Gee, 2003, Squire, 2003, Mayer, 2019), and the potential for videogames to contribute towards scientific education is highlighted in the following quote from Gee (2003, p. 20), who states that:

Designers face and largely solve an intriguing educational dilemma, one also faced by schools and workplaces: how to get people, often young people, to learn and master something that is long and challenging--and enjoy it, to boot.

As noted by Mayo (2009), videogames have tremendous mass appeal, reaching audiences in the hundreds of thousands to millions, and so videogames would seem to be an ideal medium through which to communicate geoscientific topics, especially in informal learning environments.

Ferguson, C. J. 2007. The good, the bad and the ugly: A meta-analytic review of positive and negative effects of violent video games. *Psychiatric quarterly,* 78**,** 309-316.

Gee, J. P. 2003. What video games have to teach us about learning and literacy. *Computers in Entertainment (CIE),* 1**,** 20-20.

Granic, I., Lobel, A. & Engels, R. C. 2014. The benefits of playing video games. *American psychologist,* 69**,** 66.

Griffiths, M. D. 2002. The educational benefits of videogames. *Education and health,* 20**,** 47-51.

Mayer, R. E. 2019. Computer games in education. *Annual review of psychology,* 70**,** 531-549.

Mayo, M. J. 2009. Video games: A route to large-scale STEM education? *Science,* 323**,** 79-82.

Squire, K. 2002. Cultural framing of computer/video games. *Game studies,* 2**,** 1-13.

Squire, K. 2003. Video games in education. *Int. J. Intell. Games & Simulation,* 2**,** 49-62.

Wang, B., Taylor, L. & Sun, Q. 2018. Families that play together stay together: Investigating family bonding through video games. *New Media & Society,* 20**,** 4074-4094.

*Why the 'van Gogh' filter specifically? Were other filters explored? why were they excluded? Much more detail needs to be added to explain the reasons why and the advantages/bias of using this filter over other filters.*

The van Gogh filter was chosen because it strikes a good balance between hiding the rendered nature of the images from the game, while retaining the geoscientific features. This was chosen by comparing many online filters. Although no quantitative measure for filter selection was used (we doubt one can construct one that is fair), we agree that more detail on the process will

improve the manuscript. As mentioned above in a response to reviewer #1, we suggest adding the following after P3L6:

> To make sure that all of the images were treated equally we needed a filter that works algorithmically and does not require input from a human. Many such filters exists online, yet most of them either failed in removing the rendered nature of the videogame images, or else modified the image too much, removing any the geoscientific content. The "van Gogh" filter (available through LunaPic (2015)) was chosen as it was one of the few filters which retained the geological features of the image, while hiding the rendered nature of the videogame images.

*P2 Line 7-8 "Games have a great potential for tangential learning, i.e. learning things about the real world as a tangential benefit while primarily enjoying the experience" – you state this and then immediately try to link to the potential problems (erroneous learning), but why consider the problems and not the potential benefits you reference as possible? Needs justification.*

Thank you for highlighting this, we agree that this section of the text did not flow in a particularly logical fashion. As you can see from previous comments, we have now addressed more of the potential benefits for videogames (within the scope of this study), and we will add the following text to P2L8:

> Games have a great potential for tangential learning, i.e. learning things about the real world as a tangential benefit while primarily enjoying the experience (Portnow, 2012; Mozelius et al., 2017). The tangential learning opportunities of videogames have been studied elsewhere (see e.g. Breuer et al. (2010), Mozelius et al. (2017)); however, what has not yet been fully addressed is the extent to which this tangential learning could lead to misinformation if the game world was presented in a manner that was incongruent with reality.

> Breuer, J. and Bente, G., 2010. Why so serious? On the relation of serious games and learning. *Journal for Computer Game Culture*, *4*, pp.7-24.
> Mozelius, P., Fagerström, A. and Söderquist, M., 2017. Motivating factors and tangential learning for knowledge acquisition in educational games. *Electronic Journal of e-Learning*, *15*(4), pp.343-354.

*How could you be sure that the 'fake' landscapes from the game had not been designed based on existing real-world landscapes (as surely many are)? You may not be able to evidence this, but this could be a bias in the study and should be acknowledged.*

The reviewer raises an important point. To some extent, most videogame landscapes draw inspiration from real-world landscapes, as the designers (artists) working on these landscapes often draw from their own experiences. However, in designing these landscapes, many

unrealistic elements are often added (or geoscientific laws are broken); either to add a "fantastic flair" to the landscape, or because of the limitations of being within a gameworld. For example, the rock formations in the background of the desert picture (P7 in the survey) are highly unlikely to have arisen from erosion processes, yet are very aesthetically pleasing. In the section where we discuss the potential pitfalls in interpreting the results of our study we suggest adding the following to P7L5 to acknowledge this bias:

> Landscapes in games are, of course, designed by artists who draw on the real world for inspiration. Different games will have different levels of adherence to real world inspirations, depending on the aesthetic sensibilities of the gameworld or limitations of the software used to create it. Our research focussed on the question: can non-geoscientific experts correctly identify non-realistic aspects of landscapes in videogames.

*As acknowledged by the authors, this research is in desperate need of the addition of qualitative analysis. This would also enable cognitive testing for the use of the filter and how that the use of the filter adds bias to the results, as well as supporting any hypotheses you are trying to prove/disprove.*

We agree with the reviewer that qualitative analysis would be a great follow-up to create more insight into the quantitative results from this work. See the suggested addition at the start of the reply to reviewer #2.

*P3 – why was a scaled-type question used to gather data over other types of questions? Purely for quantitative analysis? Needs justification and literature examples where possible to justify the method.*

When asking an assessment on whether someone thinks a certain photo is real or faked, there are basically two options to do so. One option is to ask "do you think this photo is real" via a yes/no question; the other is to ask "How certain are you that this photo is real" on an ordinal scale. The benefit of such ordinal scales is that they can incorporate more nuance than a simple dichotomy, as one can also indicate that one isn't sure. Such likert scales are extremely common, especially within the social sciences. Common scales are 5, 7 and 9-point scales. See Lord & Novick (1968; Statistical theories of mental test scores) or Kaplan & Saccuzzo (2009; Psychological testing) for frequently used textbooks on questionnaire design. .

*You use the differences between geoscientists with formal training and laypeople – how do you account for other bias e.g. a layperson who travels a lot or enjoys hiking in nature and is therefore more exposed to the different types of landscapes compared to somebody who spends a lot of time playing videogames inside (with the exception of BotW).*

We agree with the reviewer that the group that we identify as lay-people may be a diverse group of people and this could be an explanation for the small difference in scores between both groups. We chose to operationalize "knows a lot about geoscience" as "had formal training in the geosciences" since this allows for a clean distinction that can be clearly worded in a question. By doing this we have focussed on "do people with a formal education in the geosciences …" instead of "geoscientists". In order to make this more explicit we will add the following text to P7L8:

> Finally, the distinction between geoscientists and non-geoscientists is made based on an acknowledgment of formal education in the geosciences. In the non-geoscientists group there is potentially a large difference between people that do, and do not, get exposed to geological features; for example, through hiking, watching documentaries, etc. Follow up research, either qualitative or quantitative, could include questions on self-perceived level of geoscientific knowledge. Doing so would likely yield further information on how well versed those educated in the geosciences consider themselves, versus those not educated, perhaps demonstrating a geoscientific version of the Dunning-Kruger effect (Dunning 2011).

> David Dunning, Chapter five - The Dunning–Kruger Effect: On Being Ignorant of One's Own Ignorance, Editor(s): James M. Olson, Mark P. Zanna, Advances in Experimental Social Psychology, Academic Press, Volume 44, 2011, Pages 247-296, ISSN 0065-2601, ISBN 9780123855220, https://doi.org/10.1016/B978-0-12-385522-0.00005-6.

*P7 – How were the images distributed at the EGU? By paper on digitally? This could affect the perception of the images compared to the laypersons who saw them digitally. Simple but should be included.*

[text]
We did a small headcount and will add the following to P6L4:

> Out of the 84 experts, 34 filled in the questionnaire on paper and the rest digitally. Of the 58 non-geoscientists only 4 filled in the questionnaire on paper. Different modes of administration for both groups, could have had a small effect on the outcomes.

*Technical corrections:*
*P2 Line 14 – citation error.*

We will fix this.

*P2 Line 20-23 "However, if videogame. . ." – this sentence needs a citation.*

The following citation will be added after this sentence:

Squire, K. and Jenkins, H., 2003. Harnessing the power of games in education. Insight, 3(1), pp.5-33.

*P2 Line 30 – perhaps change rate to perceive*

We agree that this is better and will change it in the final manuscript.

*P3 Figure 1 – needs to be much larger to enable the reader to compare the quality of images.*

This is currently the standard image format dictated by the Copernicus LaTeX template. We will ask the editorial staff if this figure can be printed larger. On top of that, we suggest to add:

> High-resolution versions of images (originals and after filters were applied) are available in the supplementary material.

*P4 Line 29- 30 – this sentence is confusing – overall N=163 of which 4 and 17 participants were removed? It reads more like 163 people only completed part of the survey and were also removed! "filled out part of the survey" – maybe simply change this to 'completed the survey'.*

We will change it into 'completed the study'.

*P5 Line 6-8 – as mentioned above this could be further qualified if a qualitative approach is adopted.*

We agree with the reviewer, see our previous statements on follow-up qualitative research.

**Martin Archer**

*This is a very interesting paper on whether game environments provoke a type of 'uncanny valley' effect in their depiction of different types of landscapes and whether trained geoscientists are more effective at discerning this compared to the general public. Given that game environments are being more and more widely used to communicate geoscience, this is very relevant.*

*I have one main comment on the analysis method used and its interpretation. The C1 numerical scale participants were asked to use essentially measures two potentially independent things: whether participants identify if the picture was real (6-10) or fake (1-5), and how confident they are in that choice (the absolute distance of their score from 5.5). The penalty score used in the paper therefore is measuring a combination of both these effects. While this is appropriate for the question the authors raise of "do people without a background in the geosciences rate*

*landscapes from game worlds as more realistic compared to those with a background in the geosciences?", it is important to understand the underlying factors between the two sets of penalty scores. I would suggest the authors therefore also compare whether there are statistically significant differences in both the success rate and the confidence of geoscientists compared to non-geoscientists and whether these are independent or not. This would then determine if geoscientists are more effective or just more confident (or some combined effect) in their judgements.*

This is an interesting suggestion and we thank dr. Archer for it.
We've computes scores according to these two rubrics, the number of photos identified correctly (score between 0 and 12) and the confidence in the answers (as computed by the sum of absolute deviations from 5.5 per answer (thus maximum score = 12*4.5 = 54)). The results are as follows.

Counting number of questions correctly classified:
Geoscientists: mean = 9.524, sd = 1.793
Laypeople: mean = 9.121, sd = 1.568
Student's t-test for the difference: t = 1.385, df = 140, p = .168

Confidence in answers
Geoscientists: mean = 34.810, sd = 8.262
Laypeople: mean = 33.086, sd = 8.353
Student's t-test for the difference: t = 1.2164, df = 140, p = .226

So, to summarize: geoscientists score slightly better and are slightly more confident than laypeople, but on neither rubric the difference is significant. For this reason, we shall not elaborate upon this in the manuscript.

*In addition, it would be helpful if the authors made a short comment about the likely types of non-geoscientists they were reaching by choosing to share through their own social media channels. Recent research has highlighted the critical number of Twitter followers to start reaching outside of the scientific community [Cote and Darling, 2018, Facets, https://doi.org/10.1139/facets-2018-0002]. Beyond the findings of that study, given that the authors are geoscientists, is it not likely that the people who follow them at least have an interest in the geosciences and therefore may be performing better than a truly representative random sample of the population? If the authors have any evidence of influencers unaffiliated with geoscience who shared the survey, this would also be highly relevant to this point.*

We agree with Dr. Archer that using social media to attract attention to our survey might have introduced bias. All authors on the paper shared the survey through their twitter accounts and all have more than the lower limit of 1000 followers mentioned by Cote and Darling by at least a factor of two. We therefore will add the following paragraph after P4L10:

Advertising a survey through social media runs the risk of selecting survey participants from the biased social circle of the authors. However, Cote and Darling (2018) showed that above one thousand followers, a more diverse public is reached through twitter. Given that all the authors shared the survey through their personal twitter accounts, and that each of these accounts has more than 2,000 followers we are confident that an acceptably diverse public was reached using this approach.